# Host Specificity and Preliminary Impact of *Lepidapion argentatum* (Coleoptera, Brentidae)*,* a Biocontrol Candidate for French Broom (*Genista monspessulana*, Fabaceae)

**DOI:** 10.3390/insects12080691

**Published:** 2021-07-31

**Authors:** Elven Kerdellant, Thierry Thomann, Andy Sheppard, René F. H. Sforza

**Affiliations:** 1European Biological Control Laboratory, USDA-ARS, Campus International de Baillarguet 810, Avenue du Campus Agropolis, 34980 Montferrier-sur-Lez, France; ekerdellant@ars-ebcl.org; 2CSIRO European Laboratory, Campus International de Baillarguet, 34980 Montferrier-sur-Lez, France; Thierry.Thomann@csiro.au (T.T.); andy.Sheppard@csiro.au (A.S.)

**Keywords:** weed management, weevil, seed feeder, gall former

## Abstract

**Simple Summary:**

French broom is a leguminous shrub that is an internationally significant invasive alien weed in California and nearby US states that competes with native vegetation and increases the risk of wildfires. French broom originates from the Mediterranean region and is the target for classical biological control. Exploration for prospective biological control agents in the native region of Western Europe resulted in the collection of the weevil *Lepidapion argentatum*. We conducted preliminary field studies showing that this weevil can impact seed production in the native habitat. We also measured the specificity of this candidate biocontrol agent in laboratory conditions on 36 plant species growing in California and Australia. Our data showed that, in no-choice tests, seven species were directly negatively impacted by gall formation induced on plant stems by the weevil, but at different levels, including two lupine species commonly distributed in California. Further experiments are needed to ascertain the safety of this candidate biocontrol agent prior to release.

**Abstract:**

French broom (*Genista monspessulana*) (Fabaceae) is a perennial species native to the Mediterranean basin. Introduced in the 19th century as an ornamental plant, it is currently invasive in California and Australia. The current research is focused on biocontrol with the use of the phytophagous weevil *Lepidapion argentatum* (Brentidae). Its capacity to develop both in the stem galls and pods of French broom makes it a promising candidate. The impact on the reproduction of French broom was studied in Southern France and revealed that it could effectively reduce the number of viable seeds by 18.8%, but also increased the number of aborted seeds by 10% within the attacked pods. To evaluate the specificity of *L. argentatum*, choice and no-choice tests were performed in 2012 and 2015 on a total of 36 non-target closely related species. Results revealed the presence of galls and larvae in the stems of seven species, including two endemic Californian lupines; i.e., *Lupinus arboreus blue* and *Lupinus chamissonis*. In the future, new tests will be conducted to determine if *L. argentatum* is able to complete its entire development lifecycle on the non-target species where galls have previously been observed.

## 1. Introduction

French broom, *Genista monspessulana* (L.) LAS Johnson (Fabaceae), is a perennial woody shrub, native to the Mediterranean region commonly associated with regeneration in fire-prone environments. Introduced as an ornamental plant to many parts of the world, French broom is currently a major invader in several countries including the USA and Australia [1]. It was first introduced to California in 1871 and reported as naturalized in the 1940s [2]. Today, it is listed among the most invasive wildland plants in the state of California with infestations identified in 23 counties covering 40,000 ha [3]. In Australia, it has been determined that French broom infests at least 600,000 ha in various ecosystems including some Australian national parks [4,5,6]. The evolutionary history of French broom and its introduction is very complex with many inter- and intra-specific hybridizations in California, such as *Cytisus-racemosus* [7]. Kleist et al. [7] suggested a combination of factors (multiple introductions, hybridization between taxa) to explain the origin of the distribution observed today. In its introduced range, French broom is able to grow faster and have a longer life duration than in its native range [8,9]. In its native range, French broom has a lifetime of five to eight years and generates a seed stock of 500 to 900 seeds/m^2^/year, while, in its introduced range, it can live up to 12 years, may flower more than once a year, and produces more seeds (annual seed rain up to 5000 seeds m^2^ creating seed banks of between 30,000–100,000 m^2^) [8]. The ability of French broom to fix atmospheric nitrogen, its high growth rate, high seed production, long-lived seed bank, and the susceptibility of these disturbed ecosystems to invasion make French broom an excellent competitor, especially on poor soils. French broom tends to form dense mono-specific stands that shade out native species (e.g., endemic lupines of California). French broom also leads to an increase of the frequency and intensity of fire, and seeds are toxic for some animals (e.g., horses [3]).

Chemical and mechanical control methods are typically the first management methods used to control invasive plants. Unfortunately, chemical control of French broom is not possible because the invaded areas (especially national parks) are also home to many ecologically important protected plant species. Considering the large areas invaded by French broom, and large seed banks which allow fast regeneration, mechanical control appears difficult to achieve and expensive to apply [1,8]. Biological control of French broom remains; therefore, the most feasible strategy in a sustainable, ecological, and economic manner [10]. As a result, a classical biological control research program was initiated by Australia and California in 1999–2000 [4].

Foreign exploration in the native range (Spain, France, Italy, Greece) by collecting pods, beating tray samples, and threshing of leaves, allowed the identification of 85 species of herbivorous insects, of which 26 appear to be specific to the tribe Genisteae [11]. Of these, four species have been studied to varying degrees: *Arytinnis hakani* (Longinova) (Hemiptera, Psyllidae), *Bruchidius villosus* (F.) (Coleoptera, Chrysomelidae), *Lepidapion argentatum* (Gerstaecker) (Coleoptera, Brentidae), and *Chyliza leptogaster* (Panzer) (Diptera Psilidae). Preliminary studies have shown that the psyllid *A. hakani* had the greatest impact on French broom [8], by reducing growth, flowering [12,13], and seed production [5]. Choice and no-choice tests demonstrated that *A. hakani* can complete its development cycle on several lupine species, genetically related to French broom [14,15]. *Arytinnis hakani* was discovered in South Australia on French broom while a release permit application for it was under consideration, but risk assessment indicated low risk to the native Australian flora [4]. Thus, the next biocontrol agent candidate selected for further testing for California and Australia was *L. argentatum*. This weevil feeds on Fabaceae including *G. monspessulana* [16]. The genus, with 16 species, is native to the Mediterranean basin [17]. *Lepidapion argentatum* was considered to be monovoltine [18] overwintering as adults. Females have a dual oviposition behavior. In spring, they lay eggs inside *G. monspessulana* fresh pods and, after hatching, larvae complete their development by consuming developing seeds. However, females can also lay their eggs in young stems, inducing gall formation in which larvae complete their development to adults in 34 to 40 days at mean 23.67 ± 2.43 °C [19]. The double impact of *L. argentatum* by consuming seedpods and inducing stem galls represent a potential major asset for the biocontrol of French broom.

The major objectives of this manuscript are (1) to obtain a first assessment of the weevil’s impact on the seed set of French broom and (2) make clear the host range of *L. argentatum* by conducting host–specificity tests under controlled conditions using both choice and no-choice experimental designs with 36 non-target species, including various broom species and closely related endemic Californian lupines.

## 2. Materials and Methods

### 2.1. Evaluation of Weevil Impact on Pods

Eighty-one *G. monspessulana* pods were harvested at different heights from 10 randomly selected native plants (810 pods in total), on 24 June 2015 at St Gilles (France; N 43°39′09.66″ E 4°24′52.77″). This represented one fifth to tenth of the total pods present on each plant. For each of the ten sampled *G. monspessulana*, pods were put by group of three in Petri dishes (diameter 5.5 cm) in a growth chamber (25 °C; PP 16/8; 60% RH), until the emergence of *L. argentatum* and associated parasitoids. All insect emergence happened between 30 June, and 17 July 2015 and all weevils were counted, sexed with the sex-ratio analyzed (Chi-square test), and all parasitoids were identified and counted. All pods were dissected under a stereomicroscope (Will Strübin, magnification × 45) and categorized into (1) unattacked, with no exit hole and (2) attacked, with an exit hole indicating weevil or parasitoid emergence, and (3) attacked pods without an exit hole but showing evidence of larval development. All seeds in each pod were counted as aborted (flat thin seeds), viable or consumed seeds. All pod data were analyzed using a Kruskal–Wallis test, followed by a comparison 2–2 using the Wilcoxon test. The Chi-square test was used for proportion data (proportion of males and females emerged from pods) (R software 3.4.1, 2017).

To confirm the visual categorization of “aborted seed”, 75 seeds scored as aborted and 100 seeds scored as viable were selected from the dissected pods and subjected to germination tests. Seeds from each category were placed separately onto moistened filter paper within a Petri dish (diameter 9 cm) and incubated at 23 °C. Dishes were checked every 2–3 days over a two-month period to assess the germination rate of the “aborted” and “viable” seed categories.

### 2.2. Host–Specificity Tests

Host–specificity tests of *L. argentatum* using no-choice and choice experimental designs were undertaken in 2012 and 2015 at Montferrier-sur-Lez (France) at the Commonwealth Scientific and Industrial Research Organization (CSIRO) and at EBCL (USDA-ARS).


**a. Plant Material**


The test plant list (Appendix A) was established using the centrifugal phylogenetic approach [20], which assumes that plant species more closely related to the target weed are more likely of being attacked than more distantly related plant species. Of the original test plant list which comprised 50 species [21], 36 species were selected, mostly in the family Fabaceae, including endemic Californian lupines and crop plants (e.g., beans and peas). All seeds from the 36 selected species were sown during winters 2012 and 2015 and grown in pots. In addition, *G. monspessulana* controls were at least two-year-old plants, grown from seeds. At first, seeds were placed in a Petri dish (diameter: 9 cm) on moistened filter paper at 23 °C in the dark. Once germinated, they were transplanted into pots (13 × 13 × 13 cm) in a mixture of a quarter each of soil, peat, sand, and loam. All pots were subsequently placed in a glasshouse under semi-controlled conditions (means: 17.5 °C; 50% RH; natural sunlight) and watered two to three times per week (with an NPK fertilizer as needed).


**b. Insect Material**


*Lepidapion argentatum* adults used during host–specificity tests were field collected by beating *G. monspessulana* in spring 2012 and 2015 at St Gilles. Once collected, adults were kept in a rearing cage (85 × 50 × 80 cm with a mesh of 600 × 500 μm) containing seedlings of *G. monspessulana* in the laboratory (T: 25 °C, natural sunlight, ca. 60% RH), until they were used. After the experiments were completed, surviving individuals were re-introduced in the rearing cage. Adults remained inside the cage for a minimum of three days, and were then able to feed and eventually lay eggs on their normal host, *G. monspessulana*, before being used again for another test.


**c. Host–Specificity tests**


As females *L. argentatum* lay eggs both inside pods and stems of *G. monspessulana* (inducing galls) [19], the two contexts were assessed. The no-choice tests took place in three phases: insect confinement on plants, insect removal, and dissection of pods and/or stems.


**No-Choice Tests on Pods in 2012 and 2015**


Test potted plants were randomly positioned under a roof exposed to the open air on one side of the glasshouse building of the CSIRO European Laboratory (Temperature range: 8–30 °C; natural sunlight; ca. 60% RH). As green pods are more suitable for *L. argentatum* to oviposit in [19], branches with both flowers and young green pods were selected for each plant species to optimize relevant pod phenology for the oviposition test. Flowering shoots with young pods were then infested with five couples and covered with a dialysis tubing system (Medicell membranes Ltd., London, UK, 5 cm in diameter, and 30 cm long, *Visking Code DTV12000.13.15*), closed at both ends by a plug of foam. Each dialysis tube represents one replicate for one given plant species. After 10 to 17 days, all insects were removed and checked (alive/dead/missing). On the same day, 2 cm long pods were collected and dissected under a stereomicroscope using fine forceps and a scalpel, to identify and enumerate internal eggs or larvae. Pods under 2-cm long were discarded because they were unsuitable for oviposition.

In 2012, a total of seven lupine species were tested. Each of the four test series included at least two controls and various replicates (3 to 12 reps) of the different lupines to be tested. For the early replicates, *G. monspessulana* potted plants had no pods so commercially available *C. racemosus* potted plants were used as a replacement control as they had pods earlier and were known to be within the host range of *L. argentatum* [22]. This allowed all test plants in all replicates to be phenologically synchronized and allowed tests to cover the entire active oviposition period of *L. argentatum*.

In 2015, a total of eight plant species were tested in exactly the same manner by groups of one to three species. A total of five replications per plant were performed except for *Lupinus luteus* (three reps).


**No-Choice Tests on Stems in 2012 and 2015**


In 2012, three mating pairs of *L. argentatum* weevils were added to the dialysis tubes on each plant. Plants were covered by ventilated boxes (plastic box cut out on the top and covered with a mesh of 600 × 500 μ) and placed outdoors at the CSIRO European laboratory for 21 days. A total of eight plant species were tested, with five replicates for each species, including *G. monspessulana* controls. In 2015, potted plants were placed randomly in a growth chamber under controlled conditions (25 °C, PP 16/8; 60% RH). Plant stems were enclosed inside a dialysis tube, in the same manner as used for the no-choice tests on pods as described above, and infested with five couples for 12 days. A total of 32 species were tested in 2015 in a series of trials using five to seven nontarget plant species and five *G. monspessulana* controls per trial, and with five replicates per species except for *Baptisia australis* (two replicates) and *Cladrastis lutea* (two replicates). In both years, at the end of each test, each ventilated box and dialysis tube was removed, insects were recollected and plants placed in a glasshouse in order to continue their development. The feeding damage on leaves was visually scored (0 = no damage; 1 = 0–20% of leaves damaged; 2 = 20–50% of leaves damaged; 3 = >50% of leaves damaged) and the survival rate of *L. argentatum* was calculated. After 14 days in the glasshouse, plants were inspected under a stereomicroscope, and stems (and pods if present) were dissected to report and quantify the presence of galls and possibly *L. argentatum* larvae inside. Eggs were too difficult to observe directly.


**d. Choice Host–Specificity Tests on Lupine Stems**


Choice tests were conducted in 2015, on stems of five lupine species that were available and in a good vegetative stage. Five species of potted lupine and control (*G. monspessulana* potted plant) were randomly placed in five replicate wood-framed cages (85 × 50 × 80 cm) with side tulle walls (mesh: 600 × 500 μ) and a front wall of glass. The cages were positioned under a roof exposed to the open air on one side of the glasshouse building of the CSIRO European Laboratory (Temperature range: 8–30 °C; natural sunlight; ca. 60% RH). Two phases were followed: Phase 1 with target, and phase 2 without. In Phase 1, a total of 18 females and 15 males were introduced in the center of each cage for three days. Then, in Phase 2, *G*. *monspessulana* was removed and stored in the glasshouse and checked for galls. In addition, all of the *L. argentatum* individuals used were removed from each cage, then were immediately reintroduced in the cages only with the lupines, except for three couples per cage. These three couples were used as controls on a new *G. monspessulana* plant isolated in a smaller cage. After 10 days, all remaining living *L. argentatum* individuals were counted on each test plant species and removed from the test cages. The feeding damage was then scored (as described above) on each test plant species. All test plants were moved and kept in a glasshouse for another 14 days at which point all test and control plants were checked and stems dissected for the presence of galls which were counted.


**e. Statistical Methods**


To test differences among treatments, a Kruskal–Wallis test was used for counting data and a Chi-square test was used for proportion data (R software 3.4.1, 2017).

## 3. Results

### 3.1. Evaluation of Weevil Impact on Pods

The two categories of pods, with an exit hole and without an exit hole but with presence of larvae (Figure 1), were then pooled as infested pods and compared with uninfested pods. In total, *L. argentatum* infested around 64% of the sampled pods, with a portion of these showing signs of adult emergence, with up to five exit holes being observed. There was a total of 497 (61%) pods with an exit hole, 287 (36%) pods without an exit hole and 26 (3%) pods without an exit hole but hosting a feeding *L. argentatum* larva. After pod dissection, the different seed categories are presented in Figure 2. In pods exhibiting exit holes, there was between one and five exit holes per pod (Table 1). Following dissections, the average number of seeds per pod was 5.9 ± 0.69 (SD). Of the 4783 seeds counted, 47% were viable. The number of aborted seeds (1616; 34%) was greater than the number of seeds consumed by *L. argentatum* (902; 19%) (*p*-values = 2.2×10^−16^). For per-pod seed consumption, 33,21%, 20.61%, 7.16%, 2.59%, and 0.86% of pods dissected had one, two, three, four, or five consumed seeds, respectively. Only one pod had six seeds consumed. In total, 18.86% of all seeds were destroyed (direct impact related to seed feeding)*,* which is about a fifth of the total. The seed germination test, found zero germination for aborted seeds (n = 75), and 80% germination for viable seeds (n = 100).

The number of viable seeds in the infested pods (2.02 ± 1.33 seeds per pod) was 39% lower than in the uninfested pods (4.17 ± 1.50 seeds) (*p*-value < 2×10^−16^). Moreover, the mean number of aborted seeds was 11% higher in the infested pods (2.27 ± 1.22 seeds) than in the uninfested pods (1.51 ± 1.29 seeds) (*p*-value = 2×10^−16^) (Figure 1).

The total number of exit holes (825) exceeded the sum of *L. argentatum* and parasitoids emerged from pods (501 individuals) (Table 1). As weevils could have emerged before pod collection, the weevil sex ratio was not significantly different from 50:50 (*X*^2^ = 0.25253; df = 1; *p*-value = 0.6153).

Under field conditions, *L. argentatum* was parasitized by at least two species of hymenopteran parasitoids, a pteromalid and a braconid (exact species identification in progress). On average, more *L. argentatum* emerged than parasitoids from pods per plant (39.6 ± 22.8 and 10.5 ± 8.44, respectively) (*p*-values = 0.02699). On average, the percentage parasitism of *L. argentatum* was 22 ± 10% per plant (ranging from 5% for the plant with the lower parasitism rate, to 37%).

### 3.2. Host–Specificity Tests

In total, 2097 *L. argentatum* (1062 males and 1035 females) were collected to conduct host–specificity tests, which provided an approximate sex ratio of 1:1 (Chi-square test for difference from expected ratio of 1:1; *X*^2^ = 0.34764; df = 1; *p*-value = 0.5555).


**No-choice tests on pods in 2012 and 2015**


Larvae or eggs were only reported from pods from control *G. monspessulana plants* (Table 2). Indeed, the only eggs observed were found in the control plants (*G. monspessulana* and *C.-racemosus*) with 9.33 ± 6.03 and 8.11 ± 4.55 eggs, respectively, on average in 2012 and with 8.36 ± 10.37 eggs on average in 2015. The mean number of eggs per pod was similar between controls in 2012 (1.23 ± 0.84 and 1.32 ± 0.38) and in 2015 (1.12 ± 0.84 eggs/pods). In addition, the number of remaining living weevils observed varied from 0 to five depending on the plant species tested (Table 3).


**No-choice tests on stems in 2012 and 2015**


Of the 35 species tested, galls were observed on stems of seven species (Table 3). Two species of the genus *Genista* (*G. linifolia, G. stenopetala*), two species of the genus *Cytisus* (*C. villosus, C. proliferus*)*,* one of the genus *Spartium* (*S. junceum),* and two species of the genus *Lupinus* (*L. arboreus blue, L. chamissonis*)*,* as well as in stems of the native control *G. monspessulana* plants for each series (seeds originated from France) and those of the invasive control *G. monspessulana* (seeds originated from California, USA). Both French broom control plants exhibited the same number of larvae on stems. In addition, the number of remaining living weevils observed varied from 0 to five depending on the plant species tested (Table 3). The mean number of larvae ranged from 0.60 ± 0.89 to 14.20 ± 11.86 and most of the larvae were found alive except in *C. villosus.* Except for the two galled-lupines species, important feeding damage on leaves were observed on all of the galled species. For plant species with no galls observed, feeding damage was absent or minor, with the exception of *Lupinus albus* which presented significant damage on leaves.


**Choice tests**


As described in Table 4, 20% of stems of *G. monspessulana* (control 1) were galled during phase 1 and 87% (control 2) during phase 2. In the meantime, no galls were observed on the five tested species of lupines.

## 4. Discussion

### 4.1. Evaluation of Weevil Impact on Pods

Our preliminary results from the field-collected pod dissections indicated that the weevil infestation in Southern France reduced production of viable seeds by 18.86%. This is similar to impacts previously observed in Corsica island where *L. argentatum* consumed up to 10% of the seed production [23]. This level of impact for a seed feeder in its native range is a good starting point for considering it as a prospective biological control agent. For instance, *Coelocephalapion gandolfoi* (Coleoptera, Brentidae) is responsible for 51% of the seed damage on *Prosopis* species (Fabaceae) in the native range [24], and was selected as a candidate for the biological control of invasive *Prosopis* species in South Africa.

Pods that had at least one exit hole (i.e., minimum of one *L. argentatum* developed inside the pod) had the number of viable seeds significantly lower than that of pods with no exit hole. More than twice as many viable seeds were produced in pods with no *L. argentatum* present. Furthermore, the number of aborted seeds was significantly greater by 10.7% in pods having an exit hole or showing evidence of larval development. Although each *L. argentatum* larva only consumed one seed, feeding induced greater seed loss in the pod, through abortion of neighboring seeds (unpublished data). In addition, multiple oviposition by females in a pod appears likely to impact seed development leading to immature seed desiccation. Similar observations were made with *Apion ulicis* (Forster)(Coleoptera, Apionidae) developing in gorse pods [25]. In contrast, pods of Scotch broom (*Cytisus scoparius*) attacked by *Exapion fuscirostre* (F.)(Coleoptera, Brentidae) were found to contain fewer aborted seeds than uninfested pods [26]. In this species, Rodriguez et al. [26] suggested that the female chose pods with a smaller proportion of aborted seeds by probing the pods with their tarsi and antennae, to find seeds that are ‘more suitable’ for their progeny (Parnell 1966 in [26]). Our results and observations showed that *L. argentatum* oviposited when pods and seeds were in early development, making the discrimination of seed state difficult. Larvae appeared to feed upon the developing ovule, absorbing the nutrients destined for it and other developing ovules within the pod. This could explain why a larva feeding on only one seed could reduce a greater number of viable seeds. However, such an impact study limited to one site, one year, and a limited number of targeted plants showed its limitations and must be considered as preliminary.

*Lepidapion argentatum* also damages the host plant by inducing galls on stems [27]. There are several examples in the literature showing the negative impact of gall formers on their host plant [28,29,30,31], especially the impact of stem-galling weevils [32,33,34,35], but we found no other examples of a weevil with two modes of feeding that may reduce both seed production and growth. We could also predict that gall induction from *L. argentatum* may also impact the seed production in the following year. The double feeding guild (seeds and galls) and life history strategy of *L. argentatum* allows it to survive on (and damage) plant reproductive and non-reproductive tissues which (a) should help its establishment and (b) increase its impact potential on French broom, assuming weevil densities build up high enough following release. If *L. argentatum* shows a high degree of specificity, we believe that this weevil embodies an excellent candidate biocontrol agent for French broom.

Our data are in agreement with the recommendations of McClay and Balciunas [36] of the need to conduct a pre-release evaluation of potential impact on a target weed in order to increase confidence of potential biocontrol agent effectiveness.. Releasing ineffective agents could inadvertently cause undesirable indirect impacts through cascading trophic interactions [37,38,39].

### 4.2. Host–Specificity Tests

Our results from the no-choice test on stems indicated that larvae were present at the same frequency in the stems of the *G. monspessulana* control plants from both French and U.S. origin. *Lepidapion argentatum* gall formation occurred on seven non-target species. In both years of tests, some species were found to be unsuitable to sustain adult *L. argentatum.* Generally, less than one *L. argentatum* was recovered alive on these species. For example, when dissecting *H. macrostachya* (Fabaceae) stems, trichomes were observed that could have represented a barrier to oviposition. Five of the seven impacted tested species with galls are other brooms closely related to French broom and non-native to California and Australia. Among these, *Genista linifolia* is native to the western Mediterranean basin and became invasive in California and Australia after escaping cultivation [21]. Native to Madeira and the Canary Islands, *G. stenopetala* has naturalized in parts of the world, especially Australia, where it is considered “like a weed” [40]. *Cytisus proliferus* is native to the Canary Islands but is found in Australia and California, where it is cultivated for improving soil fertility [8,41]. *Spartium junceum* is native to the Mediterranean basin, including North Africa, Turkey, and the Middle East, and is commonly found with French broom and raises similar issues in Australia and California, where it is considered an invasive alien species [42]. *Cytisus villosus* is also native to the Mediterranean basin, but most larvae found in galls were dead, suggesting it is not a suitable host.

Live larvae and gall development were also observed on two lupine species; *Lupinus arboreus* and *L. chamissonis,* although at much lower levels than on French broom. It is important to recognize that our no-choice host–specificity tests on stems were essentially oviposition tests, and we cannot infer that the weevil was able to complete development on non-target species. Furthermore, the number of larvae (mean < 4) developed inside the stems of these lupines remained significantly lower than that observed on the target plant, *G. monspessulana* (mean 14 larvae). These results were not supported by those from choice tests on stems, where galls were not found on *L. arboreus* in all five of the replicates. Further experiments are required to verify whether or not *L. argentatum* can sustain a population across multiple generations on these two non-target species.

No-choice tests on pods showed no pod infestation on any of the 11 species tested, including seven lupine species. We also showed that *C.-racemosus* was as suitable a host as French broom for *L. argentatum,* as eggs were found in pods in the same proportion on both. *Cytisus-racemosus* is a horticultural hybrid between *G. stenopetala* and *G. canariensis,* which is not an invasive alien plant anywhere. Phylogenetically, both species are very closely related to French broom and molecular analysis showed that *G. canariensis* is even the closest *Genista* species to French broom [43].

In summary, the most conservative no-choice tests on stems identified a risk from *L. argentatum* to two non-target lupine species listed as endemic to California, by being able to initiate development in their stems (but not the pods). As this weevil has a double life cycle in both pods and stems of French broom, it is important to know if this weevil can persist only by galling stems without a need to access suitable pods. Certainly laboratory *L. argentatum* colonies on young French broom plants without pods suggested that this was possible (unpublished data). A similar equivocal result occurred during specificity tests for two other potential biological control agents of French broom: the bruchid beetle *B.s villosus* and the psyllid *A. hakani*. *Bruchidius villosus* were able to survive and lay eggs on one U.S. lupine species (*L. elegance*) [15], and *A. hakani* was able to develop on some other lupines [15]. These results led to the cessation of research on these biocontrol agents for California, pivoting this research onto *L. argentatum* (i.e., the present study).

## 5. Conclusions

The specificity tests carried out during two years in 2012 and 2015 showed that *Lepidapion argentatum,* as a gall former, is not completely specific to its host plant *G. monspessulana* being able to initiate gall development in the stems of two non-target lupine species. Its potential as a French broom biocontrol agent is still therefore in question. In order to introduce *L. argentatum* as a biological control agent into California (or indeed Australia where a few lupines are grown as crops), additional no-choice stem tests should be carried out to see if *L. argentatum* is able to survive to adulthood and maintain a population by producing multiple generations only in the stems of these two non-target lupine species. In addition, more choice tests in cages and in the field would also assist in better assessing the specificity and survivability of *L. argentatum* on key lupine non-target species.

## Figures and Tables

**Figure 1 insects-12-00691-f001:**
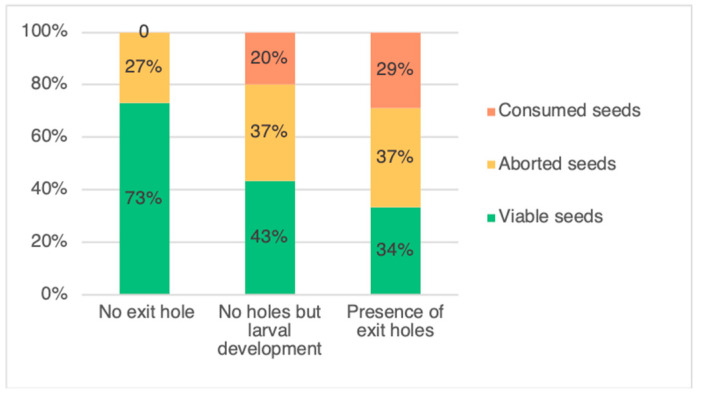
Infestation status of the three categories of pods of *Genista monspessulana*: pods with no exit hole and not infested by *Lepidapion argentatum,* pods with no exit hole but infested by *L. argentatum* (no adult emergence), and pods having at least one exit hole (adult emergence).

**Figure 2 insects-12-00691-f002:**
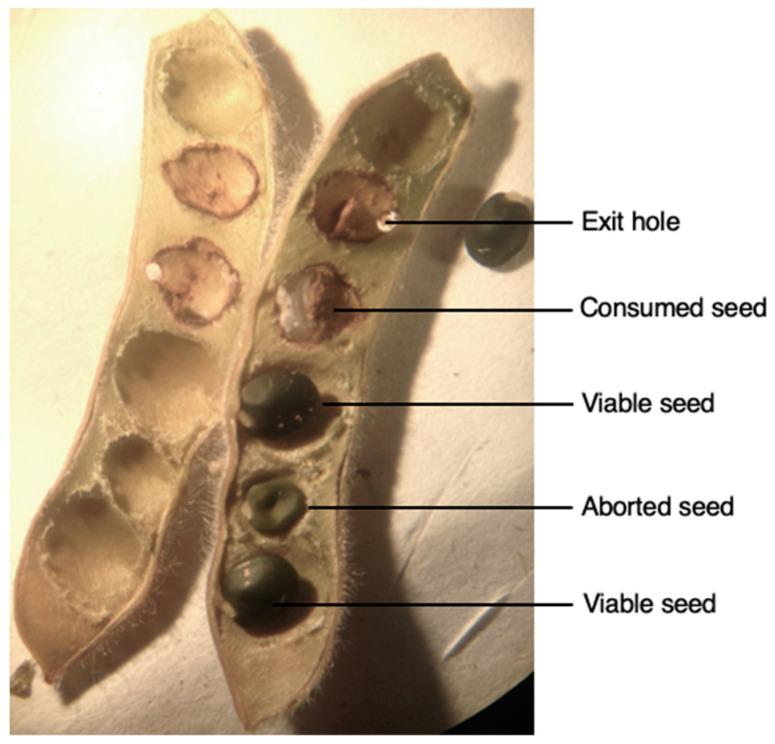
Dissection of a *Genista monspessulana* (Fabaceae) pod containing, viable, aborted and consumed seeds by *Lepidapion argentatum* (Coleoptera, Brentidae).

**Table 1 insects-12-00691-t001:** Results of *Lepidapion argentatum* adult and parasitoids emergence from randomly collected pods from 10 *Genista monspessulana* plants at St Gilles (France) in 2015.

Number of Collected Pods	Number of Exit Holes per Pod	Number of *Lepidapion argentatum* per Plant (Mean ± S.D.)		Number of Parasitoids per Plant	Parasitism Rate per Plant
	(Mean ± S.D.)	♂	♀	(Mean ± S.D.)	(Mean ± S.D.)
81 (per plant)	1.02 ± 1.06	19.3 ± 11.77	20.3 ± 11.55	10.5 ± 8.44	0.22 ± 0.10
N = 810	N = 825	N = 193	N = 203	N = 105	

**Table 2 insects-12-00691-t002:** No-choice tests on pods of test plants with *Lepidapion argentatum* including adult survival during exposure, and oviposition in pods in 2012 and 2015.

	Number of Replicates	Exposure Time (Days)	Number of Living Adults (Mean ± S.D.)		Number of Dissected Pods	Pods with Eggs/Total Pods	Number of Eggs	Number of Eggs per Pod
			♂	♀	(Mean ± S.D.)		(Mean ± S.D.)	(Mean ± S.D.)
**Species Tested in 2012**								
*Cytisus-racemosus*	9	10 to 16	4.67 ± 1.00	4.89 ± 0.33	6.22 ± 2.86	0.73	8.11 ± 4.55	1.32 ± 0.38
*Genista monspessulana*	3	12 to 14	4.67 ± 0.58	5.00 ± 0.00	8.33 ± 3.06	0.40	9.33 ± 6.03	1.23 ± 0.84
*Lupinus atlanticus consentinii*	6	13 to 18	3.67 ± 1.03	4.33 ± 0.52	6.33 ± 2.16	0	0	0
*Lupinus albus*	4	12	2.50 ± 1.73	3.75 ± 1.26	1.50 ± 0.58	0	0	0
*Lupinus angustifolius*	6	12	2.67 ± 1.37	4.50 ± 0.55	2.83 ± 1.17	0	0	0
*Lupinus consentinii*	12	12 to 18	2.58 ± 1.51	4.00 ± 1.35	2.92 ± 1.73	0	0	0
*Lupinus luteus*	4	12 to 14	2.25 ± 1.71	4.00 ± 0.82	2.25 ± 0.96	0	0	0
*Lupinus mutabilis*	8	10 to 17	1.75 ± 0.89	1.88 ± 1.36	3.13 ± 1.36	0	0	0
*Lupinus pilosus*	3	14 to 16	4.00 ± 1.00	5.00 ± 0.00	4.67 ± 1.15	0	0	0
**Species tested in 2015**								
*Genista monspessulana*	11	14	4.54 ± 0.82	4.72 ± 0.46	6.27 ± 4.10	0.66	8.36 ± 10.37	1.12 ± 0.84
*Crotalaria sagitalis* ^1^	5	14	2.20 ± 1.48	3.40 ± 2.07	2.40 ± 1.14	0	0	0
*Cytisus scoparius*	5	14	4.20 ± 0.45	3.80 ± 0.45	NA	0	0	0
*Glycine max*	5	14	0	0.60 ± 0.89	4.60 ± 1.52	0	0	0
*Lupinus albus*	5	14	3.80 ± 0.84	3.40 ± 0.89	1.20 ± 1.10	0	0	0
*Lupinus angustifolius*	5	14	5.00 ± 0.00	5.00 ± 0.00	2.00 ± 0.71	0	0	0
*Lupinus luteus*	3	14	3.40 ± 1.34	4.40 ± 0.55	2.00 ± 1.22	0	0	0
*Trifolium repens*	5	14	0	1.20 ± 1.30	NA	0	0	0

^1^ Native North American plant.

**Table 3 insects-12-00691-t003:** No-choice tests on stems of test plants with *Lepidapion argentatum* including adult survival during exposure, feeding damage, and oviposition in stems in 2012 and 2015.

	Number of Replicates	Number of Living Adults (Mean ± S.D.)		Mean of Impact Damage on Leaves *	Total Number of Exposed Stems	Number of Stems with Galls	Stems with Galls/Total Exposed Stems	Number of Galls	Number of Larvae
		♂	♀					(Mean ± S.D.)	(Mean ± S.D.)
**Species tested in 2012**									
*Genista monspessulana* (France)	5	0.80 ± 0.84	1.60 ± 1.14	2.6	n/a	NA	NA	7.80 ± 2.05	4.80 ± 0.84
*Lupinus albus*	5	0.60 ± 0.55	1.20 ± 0.84	1	n/a	0	n/a	0	0
*Lupinus angustifolius*	5	1.00 ± 0.71	1.60 ± 1.14	0.6	n/a	0	n/a	0	0
*Lupinus atlanticus consentinii*	5	0.40 ± 0.55	0.60 ± 0.55	0.6	n/a	0	n/a	0	0
*Lupinus consentinii*	5	0.60 ± 0.55	0.60 ± 0.89	0.8	n/a	0	n/a	0	0
*Lupinus luteus*	5	0.40 ± 0.55	0.80 ± 0.45	0.2	n/a	0	n/a	0	0
*Lupinus mutabilis*	5	0	0	0.4	n/a	0	n/a	0	0
*Lupinus pilosus*	5	0.40 ± 0.55	0.80 ± 0.45	0.6	n/a	0	n/a	0	0
**Species tested in 2015**									
*Genista monspessulana* (USA)	5	5.00 ± 0.00	5.00 ± 0.00	3	8.60 ± 3.78	8.00 ± 3.74	0.93	14.20 ± 11.86	14.20 ± 11.86
*Genista monspessulana* (France)	25	4.92 ± 0.28	4.76 ± 0.43	2.96	8.88 ± 3.94	8.12 ± 3.49	0.91	13.28 ± 5.43	13.08 ± 5.18
*Genista linifolia*	5	2.40 ± 0.89	4.20 ± 0.84	2.8	8.20 ± 1.92	7.40 ± 1.14	0.90	12.00 ± 1.58	12.00 ± 1.58
*Cytisus proliferus*	5	4.80 ± 0.45	5.00 ± 0.00	2.4	18.2 ± 1.48	5.20 ± 4.60	0.29	7.40 ± 7.30	7.40 ± 7.30
*Genista stenopetala*	5	4.80 ± 0.45	4.80 ± 0.45	3	11.2 ± 3.77	4.60 ± 1.82	0.41	5.60 ± 2.70	5.60 ± 2.7
*Spartium junceum*	5	5.00 ± 0.00	5.00 ± 0.00	2.2	13.4 ± 2.88	5.40 ± 2.51	0.40	6.60 ± 2.97	6.60 ± 2.97
*Cytisus villosus*	5	5.00 ± 0.00	4.60 ± 0.55	2.2	12.8 ± 2.86	1.80 ± 1.79	0.14	2.40 ± 2.07	2.40 ± 2.07
*Lupinus chamissonis* ^1^	5	3.80 ± 1.10	4.60 ± 0.55	1.2	9.80 ± 1.92	1.60 ± 3.05	0.16	3.40 ± 7.06	3.40 ± 7.06
*Lupinus arboreus blue* ^1^	5	4.60 ± 0.55	5.00 ± 0.00	1.6	8.40 ± 0.55	0.60 ± 0.89	0.07	0.60 ± 0.89	0.60 ± 0.89
*Amorpha fruticosa* ^1^	5	0	0	0	8.20 ± 1.92	0	0	0	0
*Astragalus canadensis* ^1^	5	2.20 ± 1.10	2.40 ± 1.14	0	9.20 ± 2.17	0	0	0	0
*Astragalus nuttalii* ^1^	5	0.40 ± 0.89	0.60 ± 0.89	0	7.80 ± 1.30	0	0	0	0
*Astragalus trichopodus* ^1^	5	1.00 ± 1.22	1.60 ± 2.30	0	8.40 ± 1.14	0	0	0	0
*Baptisia australis* ^1^	2	0.50 ± 0.71	3.00 ± 0.00	0	2.50 ± 2.12	0	0	0	0
*Baptisia bracteata* ^1^	5	1.40 ± 1.14	3.00 ± 1.41	0	1.00 ± 0.00	0	0	0	0
*Cladrastis lutea* ^1^	2	0	1.50 ± 2.12	1	3.50 ± 0.70	0	0	0	0
*Crotalaria sagitalis* ^1^	5	0.20 ± 0.45	4.00 ± 0.71	0.2	7.60 ± 1.14	0	0	0	0
*Cytisus scoparius*	5	4.40 ± 0.89	4.20 ± 0.84	0.6	9.80 ± 3.83	0	0	0	0
*Cytisus striatus*	5	4.40 ± 0.55	4.80 ± 0.45	0.6	14.8 ± 4.44	0	0	0	0
*Glycine max*	5	0	0.80 ± 0.84	0	4.60 ± 0.89	0	0	0	0
*Hoita macrostachya* ^1^	5	0	0	0	10.6 ± 1.14	0	0	0	0
*Lathyrus vestitus* ^1^	5	0	0.40 ± 0.55	0	5.80 ± 1.79	0	0	0	0
*Lotus scoparius* ^1^	5	0	0	0	19.2 ± 4.44	0	0	0	0
*Lupinus albus*	5	4.80 ± 0.45	5.00 ± 0.00	2.2	12.8 ± 3.49	0	0	0	0
*Lupinus angustifolius*	5	3.00 ± 0.71	4.00 ± 0.71	0.8	31.25 ± 12.12	0	0	0	0
*Lupinus luteus*	5	1.60 ± 1.82	2.80 ± 2.17	1.2	7.60 ± 3.91	0	0	0	0
*Lupinus microcarpus densiflorus* ^1^	5	0	0.40 ± 0.55	0.4	7.00 ± 1.41	0	0	0	0
*Lupinus pilosus*	5	3.00 ± 1.87	4.80 ± 0.45	0.2	9.80 ± 5.81	0	0	0	0
*Lupinus texensis* ^1^	5	4.00 ± 1.00	4.40 ± 0.55	1.4	12.4 ± 2.70	0	0	0	0
*Sesbania exaltata* ^1^	5	0.25 ± 0.45	0.25 ± 0.45	0	17.0 ± 4.64	0	0	0	0
*Thermopsis macrophylla* ^1^	5	3.40 ± 1.14	3.60 ± 1.14	0.8	6.40 ± 0.55	0	0	0	0
*Thermopsis montana* ^1^	5	4.20 ± 0.84	4.60 ± 0.55	0.8	5.80 ± 1.10	0	0	0	0
*Vicia faba*	5	0	0.60 ± 0.89	0	6.60 ± 1.14	0	0	0	0

* Damage rating on leaves 0 = no damage; 1 = minor; 2 = important; 3 = strong. ^1^ Native North American plant.

**Table 4 insects-12-00691-t004:** Choice tests on stems of test plants for *Lepidapion argentatum* including adult survival during exposure, adult feeding damage, and oviposition in stems of five lupine species and one *G. monspessulana* plant per cage exposed for three days with 18 females and 15 males (phase 1) followed by 10 days with 15 females and 12 males (phase 2).

	Number of Replicates (Cages)	Number of Adults Introduced in Each Cage (and Exposure Time)		Number of Living Adults (Mean ± S.D.)		Mean Impact Damage on Leaves *	Stems with Galls/Total Exposed Stems	Number of Galls	Number of Larvae
Species		Phase 1	Phase 2	♂	♀			(Mean ± S.D.)	(Mean ± S.D.)
*Genista monspessulana* (Control 1)	5	18 ♀ + 15 ♂ (3 days)	0	1.60 ± 1.82	2.00 ± 2.91	1.20	0.20	3.60 ± 8.05	3.60 ± 8.05
*Lupinus albus*			15 ♀ + 12 ♂ (10 days)	0.60 ± 0.89	1.80 ± 1.09	0.80	0	0	0
*Lupinus angustifolius*				0	0	0.40	0	0	0
*Lupinus arboreus blue* ^1^				0.40 ± 0.89	0.40 ± 0.89	0.20	0	0	0
*Lupinus luteus*				0.40 ± 0.89	2.00 ± 3.46	0.80	0	0	0
*Lupinus microcarpus densiflorus* ^1^				0.20 ± 0.45	1.00 ± 1.22	0.40	0	0	0
*Genista monspessulana* (Control 2)	5	0	3 ♀ + 3 ♂ (10 days)	2.20 ± 0.84	2.60 ± 0.55	2.00	0.87	15.20 ± 7.05	15.20 ± 7.05

* Damage rating on leaves 0 = no damage; 1 = minor; 2 = important; 3 = strong. ^1^ Native North American plant.

## Data Availability

Data sets can be obtained by contacting the authors.

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
