# Peer review of "Host Specificity and Preliminary Impact of Lepidapion argentatum (Coleoptera, Brentidae), a Biocontrol Candidate for French Broom (Genista monspessulana, Fabaceae)"

_insects, 2021, doi:10.3390/insects12080691_

Round 1

Reviewer 1 Report

This paper summarizes studies investigating the impact and host specificity of the weevil Lepidapion argentatum, which is considered as a potential biocontrol agent for French broom. The weevil has a curious life cycle in that it is able to develop in the pods of French broom, but also induces stem galls. This necessitates testing both phenostages of plants during host range tests.

The paper suffers from several shortcomings, which I list below.

  1. I would suggest that the native English speaker of the co-authors carefully reviews the manuscript and improves the language as necessary. I started making specific comments but then gave up because there would have been simply too many. I also don’t feel that this is the task of the reviewer.
  2. The choice of test plant species between the different test designs, unfortunately does not allow for any robust conclusions or comparisons. Please see specific comments below.
  3. Lack of some basic information makes it difficult for the reader to put the results in perspective. Please see specific comments below.
  4. The impact ‘study’ is based on data collected in one year, at one site, at one time, from 10 plants of French broom. This results in very weak data which cannot be considered representative. At least this should be clearly stated in the discussion. Also see my specific comments below.

Specific comments:

 Introduction:

In the description of L. argentatum the authors should include its average time for larval development in both pods and stem galls so that the reader can put results of host specificity tests in perspective. Also see my further comments in Results.

In addition it would be good to know whether gall induction on stems is dependent on a certain phenostage. Stem gall formers often prefer young, vegetative stems.

M&M:

I referred to the limitations of the impact study already above.

I would suggest to write host specificity tests throughout as in title and be consistent throughout the ms.

Plant material: the centrifugal phylogenetic method needs to be described in one sentence and Wapshere’s (1972) paper should be cited.

The selection of test species and the overall test list needs to be described in more detail. Please also note total number of species on the original list and how the 36 species tested in this study were selected.

Please state if/that all plants offered in tests were potted plants (vs. cut plant material).

Insect material: I guess weevils were re-introduced in the rearing cages between experiments so that they had a chance to feed on their ‘normal’ host and continue to lay eggs? If yes, please mention.

Host specificity tests: line 180: replace ‘feeding impact’ by ‘feeding damage’ and line 181: replace ‘leaves eaten’ by ‘leaves damaged’ or ‘leaves with feeding damage’

Choice tests: in which year were these tests conducted. For no-choice tests this information is given.

The test design is very hard to follow / unclear. In Table 4 you talk about two phases, but not in the text. On the other hand it sounds from the text that after 3 days controls were removed and that in a second phase, test plants were exposed to weevils without control plants present??! However, in Table 4 it does not become clear whether results refer to phase 1 or to phase 2. Since this is not separated, probably to both phases? But what’s the purpose of the two phases then?? Text and presentation of results in Table 4 need to be clarified!

How were the plants exposed in choice tests selected? The usual sequence in host specificity tests is to expose only the species which supported some development under no-choice conditions in subsequent choice tests. However, this does not seem the case. Of course practical considerations don’t always allow such a clear sequence of tests. However, this needs to be at least explained.

Results:

I don’t think Figure 1 is necessary as the information is also given in the text.

Table 1: starts by giving measurements per pod and then switches to giving measurements per plant. I don’t think the latter is very useful and would rather stick to measures per pod. In the end the Table is not really necessary either as most data is or can quite easily be given in text.

Line 262: not sure how useful it is to conduct a statistical test if neither eggs nor larvae were found on test species.

Line 263: this information (larvae per stem) should be given in the next paragraph.

No mention is made about adult survival here although it is picked up in the Discussion. Please include a sentence or two.

No-choice tests on stems: it would be important to know which instar larvae were found in test species vs controls since the discussion mentions that these were ‘essentially oviposition tests’. Does this mean that mostly L1 were found? If yes, were older instar larvae found in controls?

Line 272: Chamaecytisus proliferus but in Table 3 it says Cytisus proliferus. Be consistent.

Tables: it would be helpful if authors would mark native NA and/or Australian plants with a footnote or similar.

Discussion:  

Lines 295-97: I agree with this statement, but in order to conclude that authors would need to try extrapolating what such a level of seed reduction might mean in respect to the population dynamics of the weed. Maybe they could use the Herrera et al. 2011 paper to interfere that. In the end 20% seed reduction is not very high, but might be higher once the agent is introduced. This, plus the limitations of the study (only one site, one year, one sampling date etc.) need to be mentioned. Overall, not much can be concluded in terms of impact based on the very limited data collected.

Lines 321-323: it is true that an agent that attacks two different phenostages of the plant has advantages. However, in respect to impact it should be mentioned that a weevil only producing stem galls could also very well impact both growth and seed output.

Line 329: the authors refer to the field host range of the weevil. This would be interesting to include in more detail, preferably with a reference!

Lines 340-341: adult survival is not mentioned in results. Please add.

Lines 348-50: doesn’t that make C. proliferus a critical test species as well? Please comment.

Line 353-54: again this is not mentioned in the results section (ie.that mostly dead larvae were found in some test species)! Please add.

Line 357-58: unclear based on results why these are essentially oviposition tests when in Table 3 you give ‘number of larvae found’. Also see my previous comment.

Lines 363-64: moreover it should be tested whether the weevil also attacks the pods of these two Lupinus species.

Overall it is unfortunate that the test species that supported some gall and larval development on stems (Table 3, 2015) were not tested with pods, so no comparison is possible.

Conclusions:

Line 391: this fact needs to be mentioned earlier! E.g. when talking about the test plant list.

Supplementary materials:

I believe ‘Code’ stands for an internal acronym that could be deleted from the Table.

I would find it much more logical and in line with the phylogenetic selection process of test species to have family first, then tribe, then plant species.

CSIRO should be in capital letters.

Author Response

Please see file attached

Reviewer 2 Report

In this manuscript, the authors report on the potential of the weevil Lepidapion argentatum as a prospective agent for the biological control of the leguminous shrub, Genista monspessulana in California and Australia. The authors conducted a series of experiments to assess the impact of weevil attack on seed production under field conditions in the native range, as well as standard no-choice and choice oviposition tests to assess the host specificity of the weevil against 36 plant species.

The field study is limited in that it provides only a snap-shot in time of the impact of larval feeding on seed production. To address this in the current manuscript, perhaps the authors could provide further information on the seasonal duration of weevil larval activity compared to the annual phenology of flowering/pod production to provide a more realistic assessment of the likely impact of the weevil on overall seed reduction. See Hill, R.L., Gourlay, A.H. and Martin, L. (1991). Seasonal and geographic variation in the predation of gorse seed, Ulex europaeus L., by the seed weevil Apion ulicis Forst. New Zealand Journal of Zoology 18, 37-43.

The authors demonstrate that in no-choice tests only the target species, G. monspessulana was accepted for oviposition by Lepidapion argentatum. However, when repeated on stems, gall formation was evident on seven of the 36 plant species tested, yet no gall formation was observed when five of these species were subjected to choice-test.

Overall, the manuscript provides important information that goes a long way towards assessing the suitability of a candidate biological control agent. It is by no means novel in its approach, as it generally follows standard host specificity testing techniques, but there are no major flows that would make the manuscript unsuitable for publication in this journal.

Minor comments as follows:

Line 27. Lowercase for common names i.e. French broom.

Line 46. ‘In Australia, it has been determined that French broom infests at least 600,000 ha in various ecosystems 46 including some Australian national parks [4-6]’.

Line 66. Consider rewording sentence to, “Mechanical control is also expensive to apply over the large areas invaded by French broom, and recruitment from persistent soil seed banks allows fast regeneration”.

Line 76-77. Consider rewording the sentence to, “Preliminary studies have shown that the psyllid A. hakani negatively impacted French broom the most [8], reducing growth and flowering [12, 13] and seed production [5].

Line 86. This is a long sentence. Consider breaking it into two, by replacing the semicolon with a full stop after “Females have a dual oviposition behavior.”

Line 90. Start a new paragraph with the sentence, “The major objectives of this manuscript are to…..”

Materials and Methods

2.1. Weevil impact on pods

It’s unclear whether the eighty pods taken from each tree were pooled across all ten trees or kept separate by tree i.e. with each tree being one replicate. If the pods were pooled, please include this in the methodology, such as (Line 102) “All pods were pooled across the ten trees and then haphazardly divided into groups of three and placed into Petri dishes (diameter 5.5 cm) until the emergence of L. argentatum and associated parasitoids.” Can you also include some information on what conditions the Petri dishes were kept, such as whether they were held in controlled environment chambers (what lighting and temperatures) or at room temperature.

Line 113. Consider rewording the sentence to, “To confirm the visual categorization of “aborted seed” (flat thin seeds), 75 seeds scored as aborted and 100 seeds scored as viable were selected from the dissected pods and subjected to germination tests. Seeds from each category were placed separately onto moistened filter paper within a Petri dish (diameter 9 cm) and incubated at 23°C. Dishes were checked every 2-3 days over a two-month period to assess the germination rate of the “aborted” and “viable” seed categories.”

Line 123. Insert ‘and’ between ‘Californian lupines’, and ‘crop plants like beans and broad beans’.

Line 130. Replace ‘(13x13cmx13cm)’ with (13 x 13cm).

Line 138. Replace ‘85x50x80 cm’ with ‘85 x 50 x 80cm’

Line 150. Replace ‘eight 150 to 30° C’ with ‘8-30°C’

Line 164. Insert ‘as’ in the sentence, ‘….used as a replacement….’

Line 172. Consider rewording this sentence to, ‘Plant stems were enclosed inside a dialysis tube, in the same manner as used for the no-choice tests on pods as described above.’

Lines 173. It is a little unclear within this sentence whether the weevils were placed inside the dialysis tubing or within the cage. Consider rewording the sentence to, ‘In 2012, three mating pairs were added to the dialysis tubes on each plant. Plants were covered by ventilated boxes (plastic box cut out on the top and covered with a mesh of 600x500μ) and placed outdoor at the CSIRO European laboratory for 21 days.’

Line 178. Remove extra space for ‘25 ° C’ to ’25 °C’.

Line 181-82. There is a gap in scoring for leaves that are between 30 to 50% i.e. 2 = 6-30% of leaves eaten; 3 = >50% of leaves eaten.

Line 193. Add spaces as in (mesh: 600 x 500 μ)

Results:

3.1. Impact.

Line 211-213.’The two categories of pods, with an exit hole and without an exit hole but with presence of larvae, were pooled as infested pods (Figure 1, Figure 2) and compared with un-212 infested pods.’ But in Figures 1 and 2, these two categories are shown separately and not pooled as stated.

Figure 1 essentially repeats the data that is given within the text. If you wish to keep the figure, then summarise the key findings from the results in the text, as shown in the following example…

‘In total, L. argentatum infested around 64% of the sampled pods, with a large portion of these showing signs of adult emergence, with up to five exit holes being observed (Figure 1).’

Line 243. Should this be Figure 2, rather than Figure 1?

Line 248 and 258. There appears to be a symbol missing here for Chi.

Line 275. ‘control’ should not be in italics.

Line 277-280. Choice tests. In reporting on the gall formation and larvae per plant for G. monspessulana, only the results for Control 2 are given, not Control 1, when in fact Control 2 was really a no-choice test because the weevils “were placed on a clean G. monspessulana control plant isolated in a smaller cage”. It is unclear why there was a need to do Control 2 in first place, but I think Control 1 is the true control for the choice test between the target and the five lupine species.

Line 280. ‘5’ replace with ‘five’.

Line 298. No space between 51 %.

Line 316. This sentence needs to be reworded to, ‘Larvae appeared to feed upon the developing ovule, absorbing the nutrients destined for it and other developing ovules within the pod.’

Line 337. Please reword this sentence to, ‘Our results from the no-choice test on stems indicated that larvae were present at the same frequency in the stems of the G. monspessulana control plants from both French and U.S. origin.’

Line 366. ‘….was as suitable a host…’

Line 383. Consider rewording sentence to, ‘These results led to the cessation of research on these biocontrol agents for California…’

Line 406. Dr Michael Grodowitz

Author Response

Please see file attached

Reviewer 3 Report

it has been a pleasure to review this interesting work which evaluates the impact of the weevil Lepidapion argentatum on its target and also its host range in order to clarify if it can be a prospective biological control agent against French broom. The authors did extensive work testing more than 30 nontarget species under different conditions (choice and non-choice tests). 

However, I do not understand why the list of the species tested in the different conditions is not consistent, i.e. why those plant species that were proved to be suitable for the oviposition by the no-choice tests were not also tested in choice conditions. For example, the authors themselves benefit from that in the discussion of results obtained from L. arboreus, which was the only one tested in both conditions.

Generally speaking, I think the manuscript is well organized and it follows a clear logical path. However, in my opinion, a few points should be reviewed. For example, the issue of hybridization should be better explained since it may have important consequence in the biological control of an invasive species. The experimental design described for the choice tests should be reviewed, since some important information is missed, at least in my opinion. The results are presented not in a proper way, i.e. standard deviation or standard error are not associated with the averages presented and the p-values of some comparisons are not presented. Finally, although in a different way, the results are often presented again in the discussion, which should get straight to the point without replicating the data already presented.

My specific comments and suggestions are directly appointed in the pdf file.

My comments in details are in the pdf attached  

Author Response

Please see file attached

Round 2

Reviewer 1 Report

Please consider the additional changes below. Thank you.

Summary (line 6) : delete ‘across the insects’ life cycle’. This is not what was done since there was only one sampling date.

Plant material: reformulate addition by authors to: “The test plant list (Figure S1) was established using the centrifugal phylogenetic approach [20], which assumes that plant species more closely related to the target weed are more likely of being attacked than more distantly related plant species. Of the original test plant list which comprised 50 species [21], 36 species were selected, mostly in the family Fabaceae, including endemic Californian lupines and crop plants (e.g. beans and peas).”

Results, no-choice tests on stems in 2021 and 2015, author additions: “Both French broom control plants exhibited as many larvae on stems.” As many larvae on stems as in ??? This sentence does not make sense. Something is missing.

Discussion, 1st paragraph: I would strongly suggest to delete “Potential impact by a prospective ….” Until “… seed damage in the native range”. The first sentence is a ‘no brainer’, while the second sentence does not support the first sentence. The fact that C. ganfolfoi damages 51% of seeds of Prosopis does not prove that it can control the weed!

Discussion, next-to-last paragraph, last sentence: please include ‘we believe that’ before ‘this weevil embodies’. Based on the impact data presented I have my doubts about its impact and authors admitted that more impact data will be necessary.
